# Charged metabolite biomarkers of food intake assessed via plasma metabolomics in a population-based observational study in Japan

Eriko Shibutami[1], Ryota Ishii[2], Sei Harada[3,4], Ayako Kurihara[3], Kazuyo Kuwabara[3], Suzuka Kato[3], Miho Iida[3], Miki Akiyama[1,3,4,5], Daisuke Sugiyama[1,3,6], Akiyoshi Hirayama[4], Asako Sato[4], Kaori Amano[4], Masahiro Sugimoto[4], Tomoyoshi Soga[4,5], Masaru Tomita[4,5], Toru Takebayashi[1,3,4]*

1 Graduate School of Health Management, Keio University, Fujisawa, Kanagawa, Japan, 2 Biostatistics Unit, Clinical and Translational Research Center, Keio University Hospital, Tokyo, Japan, 3 Department of Preventive Medicine and Public Health, Keio University School of Medicine, Tokyo, Japan, 4 Institute for Advanced Biosciences, Keio University, Tsuruoka, Yamagata, Japan, 5 Faculty of Environment and Information Studies, Keio University, Fujisawa, Kanagawa, Japan, 6 Faculty of Nursing and Medical Care, Keio University, Fujisawa, Kanagawa, Japan

* ttakebayashi@keio.jp

**Data Availability Statement:** Most relevant data are within the paper and its Supporting Information files. Raw data cannot be made publicly available,

## Abstract

Food intake biomarkers can be critical tools that can be used to objectively assess dietary exposure for both epidemiological and clinical nutrition studies. While an accurate estimation of food intake is essential to unravel associations between the intake and specific health conditions, random and systematic errors affect self-reported assessments. This study aimed to clarify how habitual food intake influences the circulating plasma metabolome in a free-living Japanese regional population and to identify potential food intake biomarkers. To achieve this aim, we conducted a cross-sectional analysis as part of a large cohort study. From a baseline survey of the Tsuruoka Metabolome Cohort Study, 7,012 eligible male and female participants aged 40–69 years were chosen for this study. All data on patients' health status and dietary intake were assessed via a food frequency questionnaire, and plasma samples were obtained during an annual physical examination. Ninety-four charged plasma metabolites were measured using capillary electrophoresis mass spectrometry, by a non-targeted approach. Statistical analysis was performed using partial-least-square regression. A total of 21 plasma metabolites were likely to be associated with long-term food intake of nine food groups. In particular, the influential compounds in each food group were hydroxyproline for meat, trimethylamine-*N*-oxide for fish, choline for eggs, galactarate for dairy, cystine and betaine for soy products, threonate and galactarate for carotenoid-rich vegetables, proline betaine for fruits, quinate and trigonelline for coffee, and pipecolate for alcohol, and these were considered as prominent food intake markers in Japanese eating habits. A set of circulating plasma metabolites was identified as potential food intake biomarkers in the Japanese community-dwelling population. These results will open the way for the application of new reliable dietary assessment tools not by self-reported measurements but through objective quantification of biofluids

as study participants did not consent to have their information freely accessible. Based on this lack of consent, the Ethics Committee for Tsuruoka Metabolomics Cohort Study (which includes representatives of Tsuruoka citizens, administration of Tsuruoka City, a lawyer, and expert advisers) strictly prohibits any public data sharing because data contain potentially identifying or sensitive disease information. Data accession requests may be sent to the administration of the Ethics Committee for the Tsuruoka Metabolomics Cohort Study. The data will be shared after a review of the purpose and permission by the ethics committee. Contact information for the Ethics Committee for Tsuruoka Metabolomics Cohort Study is the administrator of the committee, Yutaka Sato, who may be contacted at the following email address: ytk.s@city.tsuruoka.yamagata.jp. Address: 9-25 Babacho, Tsuruoka City, 997-8601, Japan.

**Funding:** This work was supported in part by research funds from the Yamagata Prefectural Government (http://www.pref.yamagata.jp/) and the city of Tsuruoka (https://www.city.tsuruoka.lg.jp/) and by the Grant-in-Aid for Scientific Research (B) (grant numbers JP24390168, JP15H04778) and Grant-in-Aid for Challenging Exploratory Research (grant number 25670303) from the Japan Society for the Promotion of Science (http://www.jsps.go.jp/). The funders had no role in study design, data collection, and analysis, decision to publish, or preparation of the manuscript.

**Competing interests:** The authors have declared that no competing interests exist.

## Introduction

Nutrition studies aim to reveal associations between dietary exposure and specific health conditions by clarifying individual or group food intake. While an accurate estimation of intake is essential for accomplishing this aim, there are limitations in determining them with adequate validity and replicability. In addition to the most practical assessment tool, the food frequency questionnaire (FFQ), researchers have also utilized more effective measures, such as dietary records and 24-hour recalls [1, 2]. However, random and systematic errors affect self-reported assessments. Therefore, it is crucial to develop objective assessment tools (that is, dietary biomarkers) based on the concentrations of metabolites in biofluids such as blood and urine.

Metabolomics is one of the core subject fields of systems biology, wherein comprehensive data of all measurable metabolite concentrations are collected from biochemical samples and subjected to advanced statistical processing to derive meaningful facts [3, 4]. Also, nutrimetabolomics, which combines metabolomics and nutritional status, is an evolving field that can yield great advancements in nutrition research as a tool for objective food intake evaluations, response to nutritional modulations in observational and interventional studies, and metabolic profiles as biological consequences of dietary intake [5–9]. Advanced analytical technologies have also driven the prediction of dietary biomarkers. Capillary electrophoresis mass spectrometry (CE-MS) has enabled us to measure charged low-molecular-weight compounds with notable higher speed and resolution [10, 11] than other standard methods. Circulating blood plasma metabolites affected by habitual food intake are likely small polar molecules, including amino acids and carbohydrates, as well as their analogs and conjugates. Thus, we can expect to identify such food-specific metabolite markers comprehensively using CE-MS.

Although a considerable number of attempts have been made to identify dietary biomarkers that reflect specific food and nutrient consumption by targeted approaches, conducting non-targeted research to explore unknown full-coverage metabolites is still a fairly new approach [12–15]. Additionally, global-scale epidemiological studies have reported comprehensive investigations, focusing on the relationships between food intake, metabolites, and disease risk [16–19]; however, only a few large-scale epidemiological studies among Asian populations have so far been reported [20–22], and further research on various regional characteristics of free-living individuals is expected.

The present study aimed to clarify how habitual food intake influences circulating plasma metabolites in a free-living Japanese regional population and to identify potential biomarkers of food intake, for new reliable dietary assessment tools by objective quantification of biofluids. To achieve this aim, we conducted a cross-sectional analysis as part of a large cohort study, with charged metabolomics data obtained by CE-MS, using the partial least squares regression (PLS-R) statistical method.

## Materials and methods

### Participants and study design

The Tsuruoka Metabolome Cohort Study (TMCS) is a population-based, prospective cohort study conducted in Tsuruoka city, Yamagata Prefecture, Japan, and is designed particularly to discover metabolomics biomarkers related to environmental and genetic factors and those for common diseases and disorders. Detailed information on the cohort study methods has been published elsewhere [22–24]. Briefly, the participants of the TMCS were 11,002 residents or workers in Tsuruoka aged 34–74 years at the time of the baseline survey conducted from 2012–2015. All participants provided written informed consent for the study and its protocol was approved by the Medical Ethics Committee of the Keio University School of Medicine,

Tokyo, Japan (approval no. 20110264). Firstly, 7,303 participants aged 40–69 years without a medical history of stroke, coronary heart disease, or cancer were chosen for this cross-sectional study. Among those, the following participants were excluded from this analysis: those who did not respond to the FFQ, those who had missing data regarding staple food frequency (n = 40), those who had missing data regarding drinking status (n = 10), those with an unassessed metabolome (n = 39), outliers of biochemical test values (n = 4), outliers of estimated food frequency (n = 26), those who were not fasting before blood sample collection (n = 143), and those with missing data regarding fasting status (n = 32). Finally, 7,012 participants were included in the analysis, comprising 3,198 males and 3,814 females. A flowchart of participant inclusion and exclusion in the analysis is shown in S1 Fig.

All data and blood samples were obtained at the time of the baseline survey. The participants responded to a self-reported questionnaire that included information on demographics, physical activity, alcohol consumption, smoking habits, personal medical history, and other lifestyle factors. Energy intake and daily food consumption were assessed based on a validated FFQ (see later for details). Omissions or inconsistencies in participants' responses were addressed by the trained survey staff through face-to-face interviews. The medical history was evaluated based on both, the self-reported questionnaire and medical checkup results. Biochemical test results were obtained from medical checkup institutions with the consent of the patients, including the height and weight, to calculate the body mass index (BMI).

The fasting plasma samples were analyzed to obtain non-targeted metabolomics data, using capillary electrophoresis time-of-flight mass spectrometry (CE-TOF-MS), which predominantly measures charged low molecular compounds, such as amino acids and their analogs. A 16 ml blood sample was collected from each participant between 8:30 and 10:30 in the morning after 12 hours of fasting from the previous night to avoid short-term metabolic fluctuations. The sample was divided into 0.5–1 ml portions, then metabolites were extracted from plasma within 6 hours of collection to further minimize the effects of metabolic changes and stored frozen until used for analysis. Sample preparation methods and analysis protocols for CE-TOF-MS have been described in detail previously [22–24]. We quantified the absolute concentrations of 115 metabolites that were expected to be stably observed in most human plasma samples and were compatible for comparison with standard compounds. Raw data were analyzed with our proprietary software, MasterHands [25] (see the summary of instruments and analytical conditions in S2 Table).

## Dietary assessment

The questionnaire on dietary habits administered as part of the cohort study was created based on the Semi-Quantitative Food Frequency Questionnaire (SQFFQ) developed by the Department of Health Promotion and Preventive Medicine, Graduate School of Medicine, Nagoya City University [26, 27]. The validity and reproductivity of the SQFFQ had been assessed for energy, selected macro and micronutrients, and food consumption [28, 29]. A total of 76 questions were asked concerning the frequency of intake of 47 food items by the self-administered reminder method to assess eating habits in the past year. Responses to the questions on food intake were categorized at eight levels (never or seldom, 1 to 3 times per month, 1 to 2 times per week, 3 to 4 times per week, 5 to 6 times per week, once per day, twice per day, and three times or more per day) [27]. For staple foods such as rice, bread, and noodles, we asked about the intake frequency at breakfast, lunch, and dinner, as well as the number of portions (cups/ pieces) per serving. For alcohol, we inquired about different kinds of alcohol, the number of drinking days per month/week, and the number of drinks per occasion in a questionnaire on lifestyle (see more details of questionnaires in the S1 Table).

In the present study, the 47 food items and different kinds of alcohol assessed via questionnaires were classified into four main food categories consisting of 17 food groups [26]: energy-giving foods (rice, other carbohydrates, confectionary, and oily food), protein-rich foods (meat, fish/seafood, eggs, dairy products, and soy products), fruits and vegetables (carotenoid-rich vegetables, leafy/other vegetables, seaweed, seeds, and fruits), as well as beverages (green tea, coffee, and alcohol). The daily intake of each food group (g/d) was calculated by summing the intake of included food items. The intake of each food item was calculated by multiplying the food intake frequency (per day) by the standard portion size (in grams) set for the SQFFQ nutrition calculation. If the intake frequency was less than once per day, a conversion weight was assigned (never or seldom: 0.05, 1 to 3 times per month: 0.1, 1 to 2 times per week: 0.2, 3 to 4 times per week: 0.5, and 5 to 6 times per week: 0.8). Alcohol intake was calculated based on the reported frequency and quantity consumed per occasion. The total consumption of different kinds of alcohol was calculated according to the percentage of ethanol and shown in comparison to Japanese sake. We finally focused on the three categories (protein-rich foods, fruits and vegetables, and beverages) that were suitable for identifying food biomarkers using CE-MS. Seaweed and seeds, which had a very low intake among the target population, were excluded from the analysis.

## Statistical analysis

First, we examined the characteristics of the study population by total and sex-specific data. Data with normal distributions are reported as means and standard deviations (SDs), while skewed data are reported as medians and interquartile ranges (IQRs). For the population intake status of the 17 food groups, we calculated means and $10^{th}$-$90^{th}$ percentile ranges for both total and sex-specific data.

For metabolome data, we excluded metabolites which had plasma concentrations below the assay limit of detection (LOD) in more than 60% of the entire study population, and 94 substances (54 anions and 40 cations) were assessed in the final analysis (the list of metabolites is shown in S3 Table). For samples with undetectable levels below the LOD, values were imputed using half of the LOD values.

Firstly, a principal component analysis was performed to detect outliers, and two samples were excluded from the analysis beforehand (see details of outlier detection in S2 Fig). Since plasma metabolite concentrations are multivariate data with relatively strong correlations between substances which might change simultaneously due to biochemical interactions in vivo, we used the PLS-R model to select metabolites that greatly contributed to responses to food intake. Then, PLS-R was performed using the Nonlinear Iterative Partial Least Squares algorithm [30]. In this procedure, the intake of each food item was treated as a continuous response variable $X$, and the 94 metabolites were dealt with as continuous predictor variables $Y$. All response and predictor variables were log-transformed and standardized. For each food group, observations without responses to food intake were treated as missing values. For alcohol, only data obtained from habitual male drinkers (n = 2,449) were used.

To determine the optimal number of factors required to avoid model over-fitting, leave-one-out cross-validation (LOO-CV) was performed. Using the Van der Voet test, the optimal factor number for each food group was provided with the critical value of $p > 0.1$ for Hotelling's $T^2$ statistic. For cases in which the optimal factor was less than two, the factor number was set to two. The explained variation in the $X$ matrix ($R^2X$), the explained variation in the $Y$ matrix ($R^2Y$), the predicted variation in the $Y$ matrix ($Q^2$) and their cumulative values were calculated to confirm the goodness-of-fit of the models. In a PLS-R model, $R^2Y$ is the proportion of variance in the dependent factors that is predictable from the independent factors, while $Q^2$

is the $R^2$ when the model built on the training set is applied to the test set. Adding a factor always raises $R^2Y$, whereas $Q^2$ does not raise in case of over-fitting. Therefore, the closer the cumulative $Q^2$ is to 1, the better the predictive performance of the model. The contribution of individual metabolites in the metabolic signature for each food group were evaluated from variable importance in projection (VIP) scores and positive PLS coefficients. To further complement this, the associations between food intake and plasma metabolite levels were assessed with partial rank-order Spearman correlation coefficients, controlling for sex, total physical activity levels, and smoking status as potential confounders. Statistical analyses were performed using SAS version 9.4 (SAS Institute Inc., Cary, NC, USA), and JMP version 15 (SAS Institute Inc., Cary, NC, USA) was used for outlier detection to visualize the results.

## Results

### Participants' characteristics

Table 1 shows the characteristics of the study population. The mean age was 57.8 ± 8.2 years, and the mean BMI was 23.3 ± 3.3 kg/m$^2$, which was within the Japanese standard range (18.5–25 kg/m$^2$). The population showed general sex differences among Japanese people. That is, while males were more likely to have a BMI that fell within the overweight range, females were more likely to have a BMI that fell within the underweight range. Moreover, males were likely to have much higher current smoking and habitual drinking rates than females. Concerning nutrition status, males were more likely to have high energy intake and carbohydrate ratio and women were more likely to have a high lipid ratio.

**Table 1. Characteristics of the target population.**

| Characteristics | | Mean (SD) / Median (IQR) / Parcentage | | | | | |
|---|---|---|---|---|---|---|---|
| | | All | | Male | | Female | |
| | | n = 7,012 | | n = 3,198 | | n = 3,814 | |
| Age | years | 57.8 | (8.2)[a] | 57.7 | (8.3) | 57.9 | (8.1) |
| BMI | kg/m$^2$ | 23.3 | (3.3) | 23.9 | (3.1) | 22.8 | (3.4) |
| Energy intake | kcal/d | 1,761 | (374) | 1,974 | (386) | 1,583 | (250) |
| Alcohol intake[d] | g/d | 1.3 | (0.0–25.3)[b] | 23.9 | (2.1–47.7) | 0.0 | (0.0–2.0) |
| Total physical activity | MET・hours/w | 11.0 | (4.6–21.0) | 11.6 | (4.6–24.0) | 10.1 | (4.5–19.6) |
| Smoking | | 17.1 | %[c] | 31.9 | % | 4.7 | % |
| Ex-smoker | | 27.1 | % | 49.0 | % | 8.8 | % |
| Drinking | | 50.8 | % | 76.9 | % | 29.0 | % |
| BMI overweight | | 27.8 | % | 33.3 | % | 23.2 | % |
| BMI underweight | | 5.2 | % | 2.6 | % | 7.4 | % |
| Nutrition status: | | | | | | | |
| Protein ratio | %E | 14.0 | (1.9) | 13.5 | (1.8) | 14.3 | (1.8) |
| Fat ratio | %E | 25.3 | (6.0) | 22.4 | (5.5) | 27.7 | (5.3) |
| Carbohydrate ratio | %E | 60.7 | (7.1) | 64.0 | (6.6) | 58.0 | (6.3) |
| Total dietary fiber | g/d | 11.8 | (3.6) | 10.8 | (3.2) | 12.6 | (3.8) |
| NaCl | g/d | 9.4 | (2.1) | 9.6 | (2.2) | 9.3 | (2.1) |
| Cholesterol | mg/d | 239 | (71) | 236 | (72) | 242 | (71) |

BMI, body mass index.

[a] Mean, standard deviation in parentheses (all such values).

[b] Median, 25$^{th}$-75$^{th}$ percentiles in parentheses (all such values).

[c] Percentage for categorical variables (all such values).

[d] Values are shown as ethanol equivalent.

Table 2 shows the distribution of food classification and the mean daily intake for each food group among the population. All grouped food items are common foodstuffs that are usually eaten in a typical Japanese diet. Overall, the population had a high rice intake as staple food compared with bread and noodles, and the participants obtained more protein from fish and soy products than from meat. There were some sex differences in food intake; while males were more likely to consume higher amounts of rice and alcohol, females were more likely to consume more fruits, vegetables, and dairy products. Detailed information is shown in the S6 Table.

## Identification of food intake biomarkers

To avoid model over-fitting, we performed the LOO-CV and Van der Voet test, which was proposed as a statistical test with the $T^2$ statistic for comparing the predicted residual sum of squares from different models. The PLS-R analyses resulted in final models with a cumulative $R^2X$ range of 0.11–0.24, a cumulative $R^2Y$ range of 0.05–0.29, and a cumulative $Q^2$ range of 0.01–0.53. The food groups could be classified into three predictive performance levels: the

**Table 2. Food classification and population intake status.**

| Food group | Food item on FFQ | | Mean (10th-90th range)[a] | | | | | | | | |
|---|---|---|---|---|---|---|---|---|---|---|---|
| | | | All | | | Male | | | Female | | |
| | | | n = 7,012 | | | n = 3,198 | | | n = 3,814 | | |
| **Energy-giving foods** | | | | | | | | | | | |
| Rice | Rice | g/d | 394 | (188 - | 600) | 485 | #### | 680) | 317 | (165 - | 450) |
| Other grains/potatoes | Bread, noodles, soba, potatoes | g/d | 129 | (72 - | 204) | 131 | (68 - | 215) | 127 | (72 - | 195) |
| Confectionery | Cake, Japanese traditional sweets | g/d | 21 | (7 - | 42) | 18 | (7 - | 28) | 24 | (10 - | 47) |
| Oil | Butter, margarine, mayonnaise, oil for deep fried/stir fried | g/d | 14 | (6 - | 24) | 12 | (5 - | 22) | 15 | (6 - | 25) |
| **Protein-rich foods** | | | | | | | | | | | |
| Meat | Beef/pork, chicken, liver, ham/sausage | g/d | 41 | (17 - | 69) | 39 | (17 - | 69) | 42 | (17 - | 70) |
| Fish/seafood | Fish, shellfish, squid/shrimp/crab/octopus, fish roe, processed fish food, caned tuna | g/d | 62 | (28 - | 98) | 62 | (28 - | 100) | 62 | (27 - | 97) |
| Eggs | Eggs | g/d | 19 | (4 - | 40) | 18 | (4 - | 40) | 19 | (8 - | 40) |
| Dairy products | Milk, yogurt | g/d | 122 | (13 - | 255) | 99 | (13 - | 210) | 142 | (26 - | 255) |
| Soy products | Soybeans, tofu, fermented soy food, fried soy product | g/d | 112 | (41 - | 195) | 111 | (41 - | 195) | 113 | (42 - | 194) |
| **Fruits/vegetables** | | | | | | | | | | | |
| Carotenoid-rich vegetables | Pumpkin, carrot, broccoli, green leafy vegetables, other carotenoid-rich vegetables | g/d | 78 | (27 - | 146) | 63 | (22 - | 116) | 92 | (34 - | 166) |
| Other vegetables | Cabbage, Japanese radish, dried radish, burdock, other light vegetables, mushroom | g/d | 78 | (28 - | 140) | 61 | (24 - | 111) | 93 | (35 - | 157) |
| Seaweed | Seaweed | g/d | 2 | (1 - | 4) | 2 | (1 - | 4) | 2 | (1 - | 5) |
| Fruits | Mandarin/orange/grapefruit, other fruits | g/d | 55 | (13 - | 125) | 41 | (13 - | 89) | 66 | (17 - | 136) |
| Seeds | Peanuts/almond | g/d | 3 | (1 - | 4) | 3 | (1 - | 4) | 3 | (1 - | 4) |
| **Beverages** | | | | | | | | | | | |
| Green tea | Green tea | g/d | 230 | (11 - | 600) | 220 | (11 - | 660) | 239 | (10 - | 600) |
| Coffee | Coffee | g/d | 146 | (10 - | 300) | 134 | (10 - | 300) | 156 | (10 - | 300) |
| Alcohol[b] | Sake, beer, whiskey, wine, shochu, chuhai | g/d | 106 | (0 - | 337) | 199 | (0 - | 480) | 29 | (0 - | 93) |

FFQ, food frequency questionnaire.

[a] Values are presented as mean and 10th-90th percentiles in parentheses.

[b] Values are calculated according to the percentage of ethanol and shown in comparison to sake.

lower level for eggs ($Q^2_{cum}$ 0.02 for the final model), green tea ($Q^2_{cum}$ 0.05), and meat ($Q^2_{cum}$ 0.07); the middle level for fish/seafood ($Q^2_{cum}$ 0.21), soy products ($Q^2_{cum}$ 0.23), carotenoid-rich vegetables ($Q^2_{cum}$ 0.28), other vegetables ($Q^2_{cum}$ 0.31), and dairy ($Q^2_{cum}$ 0.33); and the higher level for fruit ($Q^2_{cum}$ 0.47), alcohol ($Q^2_{cum}$ 0.53), and coffee ($Q^2_{cum}$ 0.55). Most validation sets were fitted reasonably for studying fasting concentrations by self-reported FFQs, which are commonly reported with lower validation sets than under conditions of rapid intake. Details of the CV analyses of the goodness-of-fit are shown in S3 Fig and S4 Table.

Compounds that contributed to each food intake with high VIP are shown in Table 3. While the VIP score is generally used for screening variables with PLS modeling, the score is a relative value and has a large variation due to the variable preprocessing method. Therefore, metabolites that are shown to have predominance by univariate and/or multivariate analyses are more likely to be reliable [31]. Important metabolites were selected by referring to VIP scores and PLS coefficients as well as supplementary Spearman's correlation coefficients. Relationships among them for the characteristic food groups are illustrated in Fig 1. The correlation matrix diagram is shown in Fig 2.

Metabolites with specificities of a VIP score $\geq$ 1.5 and PLS coefficient $\geq$ 0.3 were more secure than the others as candidate compounds, considering the multiple evaluations (Fig 1). Thus, we identified a total of 21 metabolites with the criteria as candidate habitual food intake markers for nine food groups in three categories, including protein-rich food, fruit/vegetables, and beverages. Major possible markers for protein-rich food intake were hydroxyproline (VIP score 2.66) and 3-methylhistidine (3-MH; VIP score 2.11) for meat, trimethylamine-$N$-oxide (TMAO; VIP score 2.63) for fish/seafood, and choline (VIP score 2.88) for eggs. In common with these sources of animal protein, 2-aminobutyrate (2-AB) and creatine were substances that were related to changes in intake. Galactarate (VIP score 2.14), threonate (VIP score 1.97), and phenylalanine (VIP score 1.95) were marker candidates related to dairy intake. On the other hand, cystine (VIP score 1.73) and betaine (VIP score 1.53) were metabolites related to the intake of soybean and soy products. Notable metabolites common to the intake of carotenoid-rich vegetables and other vegetables were threonate (VIP scores 2.23 and 1.85, respectively) and galactarate (VIP scores 2.06 and 1.51, respectively). Also, proline betaine (VIP score 3.80) was a prominent candidate marker for the intake of fruits. For beverages, metabolites such as quinate (VIP score 4.59), trigonelline (VIP score 3.13), and hippurate (VIP score 1.88) showed a relation with coffee consumption. Pipecolate (VIP score 2.78), and 2-AB (VIP score 1.92) were closely related to alcohol metabolite concentrations (see more details of the results in the S5 Table).

## Discussion

A lot of pioneering efforts of dietary biomarkers have been reported so far, aiming at several applications, such as objective quantification of specific metabolites related to food intake [12, 15], identification of proper dietary patterns by interventions [13, 14, 17, 32], and dietary profiling in epidemiological studies [32, 33]. Furthermore, large-scale metabolomics studies across a wide range of countries and regions have reported extensive investigations, to clarify the relationships between food intake, metabolites, and disease risk. For instance, the International Study of Macronutrients and Micronutrients and Blood Pressure (INTERMAP) [16, 18, 34, 35] reported that significant relationship of metabolic profiles associated with diet, xenobiotics and blood pressure levels among populations in UK, US, China and Japan, whereas the European Prospective Investigation into Cancer and Nutrition (EPIC) [17, 19, 36, 37] has revealed that the metabolic signatures were affected by specific food consumption, such as meat, alcohol, and coffee, through the dietary assessments across four European countries. For

**Table 3. Promising food biomarker candidates (n = 7,012).**

| Food group | Metabolite | Sub Class[a] | PLS-R[b] | | | $r_s$[d] |
|---|---|---|---|---|---|---|
| | | | VIP | Coeff | $Q^2_{cum}$[c] | |
| Meat | | | | | | |
| | Hydroxyproline | AA | 2.66 | 0.07 | 0.07 | 0.09 |
| | 3-Methylhistidine | AA | 2.11 | 0.06 | | 0.08 |
| | beta-Alanine | AA | 2.05 | 0.05 | | 0.04 |
| | 2-Aminobutyrate | AA | 2.01 | 0.05 | | 0.05 |
| | Creatine | AA | 1.99 | 0.06 | | 0.05 |
| | Carnitine | AA | 1.7 | 0.04 | | 0.03 |
| Fish/seafood | | | | | | |
| | Creatine | AA | 3.19 | 0.1 | 0.21 | 0.18 |
| | Trimethylamine-N-oxide | AO | 2.63 | 0.09 | | 0.15 |
| | Cystine | AA | 2.26 | 0.07 | | 0.12 |
| | 2-Hydroxybutyrate | AA | 1.73 | 0.04 | | 0.11 |
| | Isethionate | AHA | 1.55 | 0.03 | | 0.08 |
| | Glucuronate | CHO | 1.43 | 0.04 | | 0.13 |
| | 2-Aminobutyrate | AA | 1.36 | 0.03 | | 0.07 |
| | Uridine | PN | 1.32 | 0.03 | | 0.06 |
| | Guanidinosuccinate | AA | 1.21 | 0.02 | | 0.07 |
| Eggs | | | | | | |
| | Choline | QA | 2.88 | 0.05 | 0.01 | 0.06 |
| | 2-Aminobutyrate | AA | 2.4 | 0.04 | -0.02 | 0.04 |
| | Betaine | AA | 2.14 | 0.04 | | 0.05 |
| | Asparagine | AA | 1.66 | 0.02 | | 0.02 |
| Dairy | | | | | | |
| | Galactarate | CHO | 2.14 | 0.08 | 0.33 | 0.09 |
| | Threonate | CHO | 1.97 | 0.07 | | 0.09 |
| | Phenylalanine | AA | 1.95 | 0.08 | | 0.08 |
| | Lysine | AA | 1.6 | 0.04 | | 0.05 |
| | Tyrosine | AA | 1.53 | 0.04 | | 0.02 |
| | Citrate | TCA | 1.47 | 0.07 | | 0.07 |
| | Tryptophan | AA | 1.44 | 0.02 | | 0.03 |
| | 2-Aminobutyrate | AA | 1.31 | 0.05 | | 0.07 |
| | Hippurate | BA | 1.27 | 0.05 | | 0.08 |
| | Creatine | AA | 1.24 | 0.03 | | 0.02 |
| Soy products | | | | | | |
| | Cystine | AA | 1.73 | 0.07 | 0.23 | 0.08 |
| | Betaine | AA | 1.53 | 0.06 | | 0.07 |
| | Isethionate | TCA | 1.34 | 0.02 | | 0.09 |
| | Creatine | AA | 1.34 | 0.05 | | 0.08 |
| | Uridine | PN | 1.3 | 0.04 | | 0.06 |
| | Citrate | AA | 1.25 | 0.04 | | 0.06 |
| | Phenylalanine | AA | 1.25 | 0.03 | | -0.02 |
| | Glutamine | AA | 1.25 | 0.04 | | 0.05 |
| | - | - | - | - | | - |
| | - | - | - | - | | - |
| | - | - | - | - | | - |
| Food group | Metabolite | Sub Class[a] | PLS-R[b] | | | $r_s$[d] |
| | | | Carotenoide-rich vegetables | | | |
| | Threonate | CHO | 2.23 | 0.07 | 0.28 | 0.09 |
| | Galactarate | CHO | 2.06 | 0.06 | | 0.07 |
| | Creatine | AA | 1.8 | 0.06 | | 0.05 |
| | Lysine | AA | 1.44 | 0.02 | | 0.03 |
| | Cystine | AA | 1.4 | 0.04 | | 0.07 |
| | Citrate | TCA | 1.33 | 0.04 | | 0.06 |
| | Hippurate | BA | 1.29 | 0.04 | | 0.07 |
| Other vegetables | | | | | | |

*(Continued)*

**Table 3.** (Continued)

|  |  |  |  |  |  |  |
|---|---|---|---|---|---|---|
|  | Creatine | AA | 2 | 0.07 | 0.31 | 0.05 |
|  | Threonate | CH | 1.85 | 0.05 |  | 0.06 |
|  | Galactarate | CH | 1.51 | 0.04 |  | 0.02 |
|  | Cystine | AA | 1.4 | 0.04 |  | 0.06 |
| **Fruits** |  |  |  |  |  |  |
|  | Proline betaine | AA | 3.8 | 0.23 | 0.47 | 0.27 |
|  | Threonate | CHO | 2.3 | 0.09 |  | 0.15 |
|  | Galactarate | CHO | 1.95 | 0.07 |  | 0.11 |
|  | Tyrosine | AA | 1.49 | 0.03 |  | 0 |
|  | Lysine | AA | 1.43 | 0.02 |  | 0.03 |
|  | Cystine | AA | 1.29 | 0.04 |  | 0.06 |
|  | Creatine | AA | 1.29 | 0.06 |  | 0.04 |
|  | Citrate | TCA | 1.21 | 0.05 |  | 0.06 |
| **Green tea** |  |  |  |  |  |  |
|  | Threonate | CHO | 3.54 | 0.06 | 0.05 | 0.11 |
|  | Galactarate | CHO | 3.15 | 0.06 |  | 0.08 |
|  | Cystine | AA | 1.93 | 0.04 |  | 0.07 |
|  | Creatine | AA | 1.87 | 0.03 |  | 0.06 |
|  | 2-Aminobutyrate | AA | 1.74 | 0.03 |  | 0.06 |
|  | Trimethylamine-*N*-oxide | AO | 1.71 | 0.03 |  | 0.07 |
|  | Proline betaine | AA | 1.68 | 0.03 |  | 0.05 |
|  | 2-Hydroxybutyrate | AA | 1.29 | 0.02 |  | 0.06 |
| **Coffee** |  |  |  |  |  |  |
|  | Quinate | ALC | 4.59 | 0.29 | 0.55 | 0.39 |
|  | Trigonelline | AL | 3.13 | 0.17 |  | 0.28 |
|  | Hippurate | BA | 1.88 | 0.07 |  | 0.17 |
|  | Leucine | AA | 1.34 | 0.02 |  | 0.01 |
| **Alcohol[e]** |  |  |  |  |  |  |
|  | Pipecolate | AA | 2.78 | 0.17 | 0.53 | 0.26 |
|  | 2-Aminobutyrate | AA | 1.92 | 0.12 |  | 0.17 |
|  | Choline | QA | 1.87 | 0.09 |  | 0.15 |
|  | Threonine | AA | 1.65 | 0.09 |  | 0.1 |
|  | Carnitine | AA | 1.41 | 0.07 |  | 0.09 |
|  | Tyrosine | AA | 1.34 | 0.06 |  | 0.08 |
|  | Malate | BHA | 1.3 | 0.08 |  | 0.14 |
|  | Creatine | AA | 1.24 | 0.04 |  | 0.09 |

PLS-R, partial least square regression; VIP, variable importance in projection; AA, amino acids, peptides, and analogs; CHO, carbohydrates and carbohydrate conjugates; AO, aminoxides; AHA, alpha-hydroxy acids and derivatives; PN, pyrimidine nucleosides; QA, quaternary ammonium salts; TCA, tricarboxylic acids and derivatives; BA, benzoic acids and derivatives; ALC, alcohols and polyols, and polyols; BHA, beta-hydroxy acids and derivatives.

[a] Reference: The Human Metabolome Database (https://hmdb.ca).

[b] Metabolites which indicate VIP scores ≥ 1.2 and positive PLS coefficients ≥ 0.02 are shown.

[c] Cumulative predicted variation in the *Y* matrix for optimal factor numbers, calculated as 1 −(the cumulative predicted residual sum of squares / the cumulative sum of squares). The value indicates the predictive performance of the model. For cases with an optimal factor number of less than two, the factor number was set to two and the result was shown in parentheses.

[d] Partial rank-order Spearman's correlation coefficients between food consumption and metabolite concentration, controlling for sex, smoking, and physical activity levels.

[e] Data of male drinkers (n = 2,449) were used in the analysis.

population profiling of dietary habits, it is essential to accumulate results from diverse groups to consider such geographical differences. However, these pioneering efforts often assessed the effects of short-term intake by intervention trials, or were mostly implemented among Western populations.

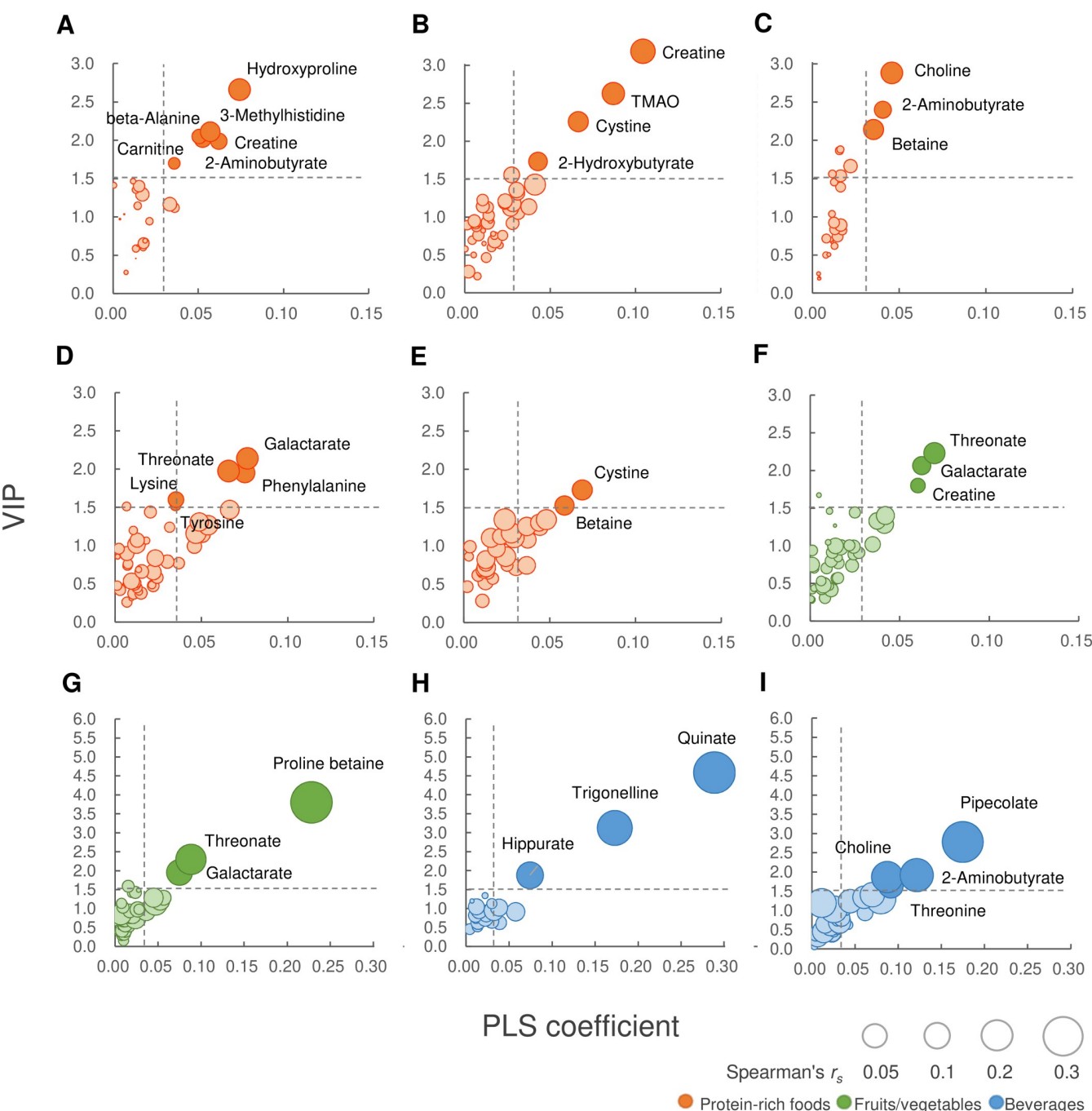

**Fig 1. Overview of food biomarker candidates assessed by PLS-R (n = 7,012).** Relationships between the VIP score and the PLS coefficient are described using Spearman's correlation coefficients. The vertical axis corresponds to the VIP score, the horizontal axis corresponds to the PLS coefficient, and the area of each circle corresponds to the correlation coefficient. Notable metabolites with VIP scores ≥ 1.5 and PLS coefficients ≥ 0.03 are highlighted. (A) meat, (B) fish/seafood, (C) eggs, (D) dairy, (E) soy products, (F) carotenoid-rich vegetables, (G) fruits, (H) coffee, and (I) alcohol. PLS-R, partial least square regression; VIP, variable importance in projection; TMAO, trimethylamine-*N*-oxide.

To the best of our knowledge, this is the first result of a comprehensive search for the effects of dietary intake on plasma metabolite concentrations in a free-living Japanese population. Twenty-one metabolites were identified as candidate habitual dietary markers for nine food

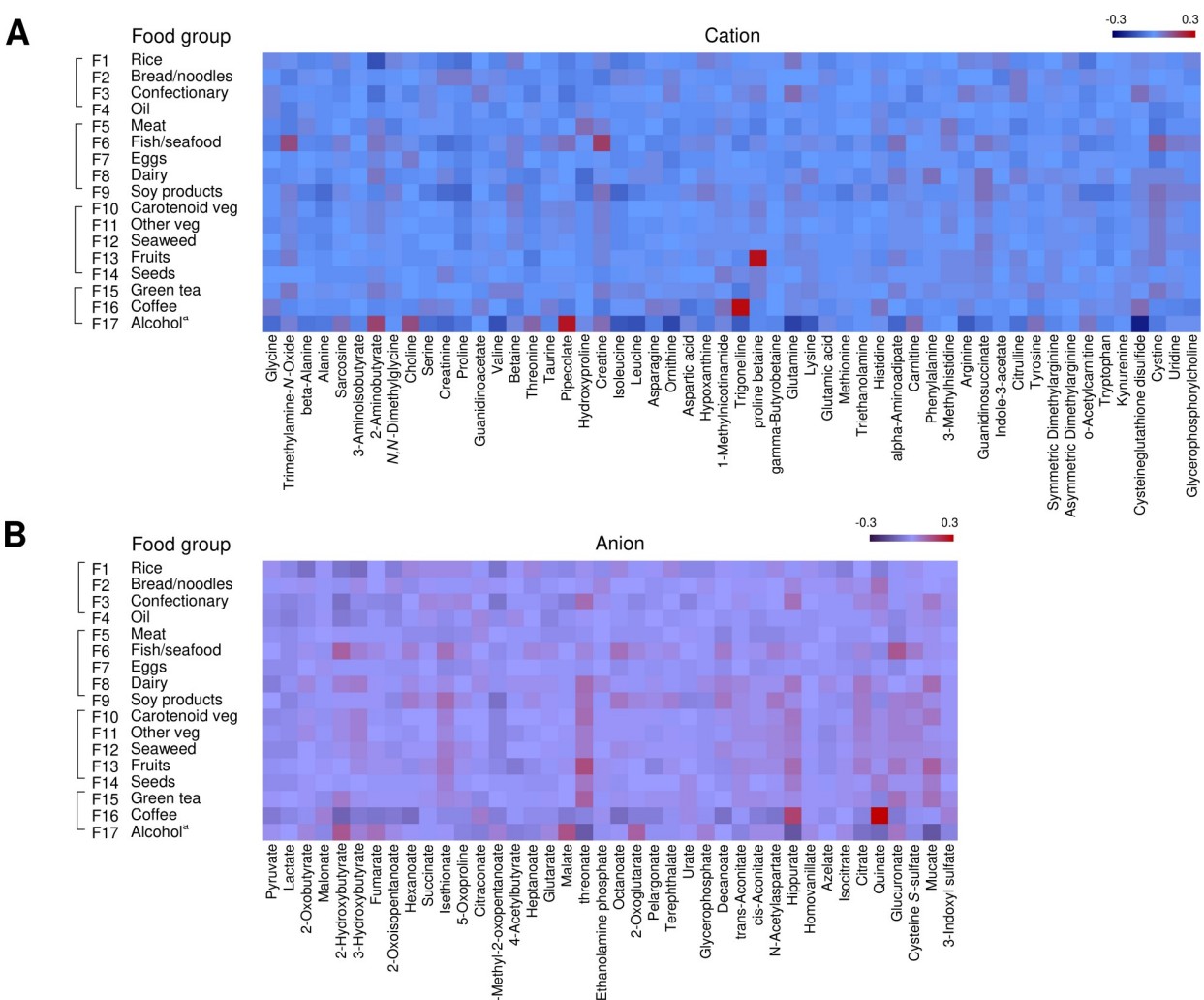

**Fig 2. Correlation matrix diagram of the relationship between food groups and metabolites (n = 7,012).** This heatmap was generated with partial rank-order Spearman correlation coefficients, controlling for sex, smoking, and total physical activity levels. (A) cation, (B) anion. [a] Data of male drinkers (n = 2,449) were used in the analysis for alcohol.

groups in three categories, as mentioned above. Most of the results were consistent with those of previous studies conducted in Western countries, while some likely reflected the characteristics of Japanese eating habits. These results demonstrated that dietary assessment was feasible for assessing long-term habitual intake by plasma metabolome analysis. The metabolites were likely exogenous dietary metabolites rather than endogenous metabolites. Since these disease markers indicated possible correlations with the intake of specific foods, the results can be useful references for considering the effects of dietary factors in further studies on disease biomarkers.

## Protein-rich foods

**Meat.** Meat intake biomarkers are important because meat consumption is likely to be associated with the risk of various cancers and chronic diseases [38, 39]. Hydroxyproline, 3-MH, and beta-alanine were identified as metabolites specific to meat consumption. Hydroxyproline is a metabolite of the nonessential amino acid proline and is mostly found in

the collagen in animal tissue. Circulating plasma hydroxyproline is mainly derived from the diet, while some are synthesized from glutamate, arginine, and ornithine [40]. Proteins from meat, poultry, and fish rather than plant-based proteins have been reported as great sources of proline and hydroxyproline [41]. Other epidemiological studies have also identified the metabolites as potential biomarkers of meat consumption [42, 43].

3-MH is most often found in the muscle of animals; thus, its dietary sources are likely to be mainly animal protein sources, such as meat and fish [42, 44–46]. Some studies have reported that whereas 1-methylhistidine (1-MH) is more likely to reflect dietary factors, 3-MH concentrations in urine and plasma tend to reflect muscle catabolism and muscle mass [47, 48]. Meanwhile, other studies have indicated that excretion of both 1-MH and 3-MH into the urine increases after meat intake [42, 45]. The cross-sectional EPIC study has also reported the specificity of 3-MH in urine for poultry intake [37].

Similarly, beta-alanine, a structural isomer of alanine, is not a protein constituent amino acid but a component of dipeptides of carnosine and anserine, most of which are present in skeletal muscle [49]. Beta-alanine intake in the diet is higher when consuming animal protein foods than consuming plant-based foods; thus, the metabolite has been considered as a potential biomarker for the consumption of meat, particularly red meat [43].

Also, our result showed that creatine was a potential contributor to the discriminability of animal protein intakes, such as meat and fish consumption. Plasma and urinary concentrations of creatine and creatinine are generally used as biochemical indicators of health statuses, such as renal function and muscle mass [50], whereas relatively high amounts of the components are also found in dietary sources such as meat and fish [51, 52]. Dietary profiling based on concentrations of creatine and creatinine in biofluids has already been reported in comparing dietary patterns between different populations [34, 44–46].

**Fish/Seafood.** TMAO concentrations increased with the intake of fish/seafood. TMAO is a non-protein amino acid that relates to the function of regulating osmotic pressure in fish. Several studies have already identified such an association between fish/seafood intake and TMAO concentrations in plasma and urine samples [37, 48, 53]. Incidentally, some other studies have revealed that plasma TMAO concentrations are positively associated with cardiovascular disease (CVD); hence, the metabolite is likely to be regarded as a potential CVD risk marker mainly among Western meat-eaters [54–56]. These studies explained that the underlying mechanism involved the metabolism of choline and carnitine contained in foods such as meat, eggs, and dairy products to trimethylamine (TMA) by gut microbiota and further metabolism to TMAO in the human liver. Then, the circulating plasma TMAO in the vessels promotes the up-regulation of macrophage scavenger receptors in the vessel, which are involved in atherosclerosis [56]. As is the case for a region with a high fish intake like our study area, however, the increase in the plasma concentrations of TMAO was likely to be based on a diet containing free TMAO from seafood. As the typical inverse relationship between fish intake and CVD risk [57, 58] contradicts the utility of TMAO concentrations as a high-risk marker, the above risk supposition might need to be modified for populations with high fish intake. Incidentally, the result of this study did not show a particular relationship between TMAO concentrations and meat and/or egg consumption.

**Other protein-rich foods.** Choline showed a possible relationship with increasing egg intake. Indeed, egg yolk is a known rich dietary source of choline and choline phospholipids [59, 60]. The major metabolites related to the consumption of dairy products were galactarate, threonate, and phenylalanine. Galactarate, a sugar acid, was likely formed from the oxidation of plasma galactose, which is a monosaccharide decomposed from lactose abundant in dairy products. Studies have revealed that urinary and plasma concentrations of galactarate were elevated in healthy adults after dairy consumption [61, 62]. Threonate is also a sugar acid derived

from the oxidation of threose (a pentose) and has also been reported as a metabolite of ascorbic acid [63]. Phenylalanine, an essential amino acid, is a natural component of the breast milk of mammals [64].

Cystine and betaine were metabolites related to the intake of soybean products rich in vegetable proteins. Cystine is a dimeric nonessential amino acid formed by the oxidation of cysteine and is abundant in soybeans as well as many other foods. On the other hand, while raw soybeans contain a large amount of choline, which is a precursor of betaine, tofu, a soybean product, tends to contain more betaine [59]. Choline, betaine, and methionine are involved in betaine metabolism in vivo [60], and betaine is a component that enhances the rotation of the remethylation pathway of methionine metabolism [65]. Therefore, the metabolic pathway of methionine-homocysteine-cysteine, whose antagonism is a risk factor for various diseases, may likely be smoothly accelerated among soybean consumers.

An increase in 2-AB concentrations was related to the consumption of foods rich in animal proteins such as meat, fish, and eggs, consistent with the results of previous epidemiological studies [22, 34, 46]. However, it might result from endogenous metabolic changes caused by the daily overconsumption of foods and drinks that accompany high meat intake, and not a direct influence of dietary animal protein components. 2-AB is a known intermediate metabolite derived from the catabolism of the essential amino acids methionine, threonine, and serine, possibly influenced by a change in the pathway of hepatic glutathione metabolism [66]. 2-AB has been reported as a biomarker associated with abnormal amino acid metabolism, likely leading to chronic alcoholic and/or nonalcoholic liver disease caused by various lifestyle-related diseases [67]. As described later, the plasma level of 2-AB also increased highly with alcohol consumption in the present study.

### Fruits and vegetables

Metabolites that were common to fruits and vegetables were threonate and galactarate, consistent with the results of previous studies [14, 32, 34]. As mentioned above, threonate, a sugar acid of threose, is also disassembled from ascorbic acid. It is generally known that fruits and vegetables are rich in ascorbic acid [63]; thus, the component was likely to influence the changes in threonate concentrations. Besides, the plasma concentrations of galactarate (mucate), a sugar acid of galactose, also increased with fruit and vegetable consumption. The metabolite is found in many foods that contain mucins, such as vegetables, potatoes, and root vegetables, as well as ripe fruits that are high in pectin (such as pear, peach, and pomes). Pectin is a structural acidic heteropolysaccharide present in the primary cell walls of plants, in which mucate exists in the form of the polymer polygalacturonate [68].

A metabolite closely related to fruit intake was proline betaine, which is known to be one of the most secure food biomarkers in plasma and urine, particularly for the consumption of citrus fruits such as oranges and grapefruits [8, 35, 69]. The metabolite is a rich component of citrus fruits and so may serve as an indicator of a healthy diet as an intake marker. Mandarin oranges are one of the most commonly consumed fruits in Japan, and the FFQ results also confirmed that citrus fruits including mandarin were consumed in large amounts in the survey population. Therefore, proline betaine was shown to be a potential biomarker for citrus fruit consumption in the Japanese population as well.

### Beverages

Three substances (quinate, trigonelline, and hippurate) were prominent metabolites related to coffee consumption, consistent with previous reports [36, 70, 71]. Esters of quinate and caffeate are abundant in coffee beans in the form of chlorogenate, which is the most commonly

known coffee polyphenol and easily pyrolytically decomposed into these two compounds by heating [72]. Besides, trigonelline is a methyl betaine of nicotinate (niacin), which is also found in high levels in coffee beans [36]. Hippurate is known to exist in the urine of herbivores, and it has also been reported that it is biosynthesized from quinate by the gut microbiota [73].

Endogenous metabolites such as pipecolate and 2-AB, which are likely to indicate chronic metabolic changes, were shown to be markers of alcohol consumption. Plasma amino acid abnormalities have been frequently reported in alcoholics. Pipecolate is a metabolite of the essential amino acid lysine, generally found in urine and plasma. Studies have shown that plasma concentrations of pipecolate are elevated in patients with chronic liver diseases [74]. Besides, as mentioned earlier, the higher plasma concentrations of 2-AB with habitual alcohol intake may reflect altered glutathione metabolism and lipid peroxidation due to alcoholic liver dysfunction [66]. Indeed, comparative studies of healthy populations have suggested that active drinkers without the liver disease have higher 2-AB concentrations than non-drinkers [22, 75].

We avoided identifying potential markers for green tea consumption. The intake of green tea, the most common beverage consumed while eating in Japan, tends to increase with the frequency of meals in Japan. Therefore, metabolites with high VIP scores such as threonate, galactarate, proline betaine, cystine, and TMAO might reflect the relations with Japanese dietary patterns that are rich in fish and vegetables, rather than specific for green tea itself. Also, although tea catechins were likely to be characteristic components of green tea [76], they were not suitable for measurement via CE-MS because they are non-polar high-molecular-weight polyphenols.

## Strengths and limitations

The present study had several strengths that should be noted. Being part of a large cohort study, there were enough data to draw a statistically supported and meaningful conclusion on the behavior of a large population. The study was also carefully designed for both epidemiological and metabolomics analyses. This indicates that there was an advantage in identifying circulating blood metabolites in long-term dietary habits in free life, which was carried out with minimal metabolic variations under a strict protocol by a non-targeted approach. Moreover, the study was carefully executed under overnight fasting conditions to limit the short-term effects of food intake.

Also, our study was pioneer research that aimed to clarify candidate food biomarkers in the Asian population, particularly habitual Japanese characteristics, although its findings may have limited generalizability. The study area was a rural town chosen for its steady population that remained less affected by rapid Westernization, unlike other urban areas. Therefore, it was ideal for research on long-term eating habits. Our findings will also help researchers to consider the influence of dietary factors in exploring biomarkers in various fields.

Despite these strengths, the study had some limitations. While large-scale epidemiological studies such as the cohort study on which our cross-sectional study was based have the great advantage of deriving meaningful knowledge from a large amount of data, it was necessary to adopt practical methods for collecting such data. Dietary assessment by FFQ, which represents participants' habitual dietary intake over a longer time, is generally regarded as advantageous in terms of cost and time. However, the method tends to have inferior accuracy randomly as well as lower estimates of intake systematically. Having said this, in the statistical analysis of the present study, a relative log-normalized amount of each food was used instead of the absolute amount. Thus, it may have been affected by random errors rather than systematic errors.

An assessment categorized by food groups is likely to make it more challenging to distinguish the effects of specific food items that may differentially associate with metabolites, but it may also improve the interpretability of dietary status based on complex intake of various food items. Thus, we can say it is suitable for practical applications to assess long-term dietary habits. Indeed, the approach by food groups has been adopted to identify habitual food intake biomarkers in typical epidemiological studies [14, 64, 71].

Although CE-MS is an optimal metabolomics measurement technology with high resolution for capturing intracellular metabolites, it was unsuitable for measuring the levels of low polar molecules like lipids and high-molecular-weight polyphenols, which are associated with the intake of various foods. It is difficult to completely cover the metabolite profile with one platform; thus, dietary biomarker identification consisting of a wider range of chemical classes may need to be assessed with an integrated approach.

Another important question is whether the metabolite signatures detected statistically here could be used as dietary biomarkers. Among various types of dietary biomarkers defined previously [8, 9], it is arguable whether the metabolite signatures we observed could be defined as biomarkers of intake (*i.e.*, concentration of replacement dietary biomarker). Follow-up surveys are ongoing now to estimate diet-disease risk association as well as to examine the function, mechanism of action, and the validity and reproducibility of the objective quantities. Effects of endogenous and extrinsic factors on metabolite concentrations are also to be evaluated.

Finally, dietary biomarkers are influenced not only by the intake of individual foods but also by the interaction of sex-specific and/or personal dietary behaviors and preferences. Therefore, it is important to consider such other factors in future studies of dietary biomarkers. The effects of gut microbiota and genetic factors on a diet also cannot be overlooked.

## Conclusions

In conclusion, a total of 21 metabolites were identified as potential habitual dietary biomarkers for nine food groups in a Japanese community-dwelling population. In particular, hydroxyproline for meat, TMAO for fish, choline for eggs, galactarate for dairy, cystine and betaine for soy products, threonate and galactarate for carotenoid-rich vegetables, proline betaine for fruit, quinate for coffee, and pipecolate for alcohol were considered as prominent food intake markers of Japanese eating habits. These results will open the way for the application of new reliable dietary assessment tools by objective quantification of biofluids. Our findings will also help to consider the influence of dietary factors in exploring biomarkers in various fields.

## Supporting information

**S1 Fig. Flow diagram of participant inclusion and exclusion in the present study.**
(PDF)

**S2 Fig. Outlier detection analysis.** Principal component (PC) analysis was executed for detecting outliers and excluded two samples from the analysis beforehand. (A) 3-D score plots for PC1, PC2, and PC3, (B) 2-D scatter plots of scores and loadings for PC1, PC2, and PC3, and (C) $T^2$ statistics for 12 principal components. UCL, upper control limit.
(PDF)

**S3 Fig. Cross-validation analyses for the goodness-of-fit of models.** As we applied Van der Voet's test to avoid over-fitting, the number of factors extracted was the lowest with residuals that were insignificantly larger than the residuals of the model with the minimum predicted residual sum of squares (PRESS). (A) meat, (B) fish/seafood, (C) eggs, (D) dairy, (E) soy products, (F) carotenoid-rich vegetables, (G) other vegetables, (H) fruits, (I) coffee, (J) green tea,

and (K) alcohol.
(PDF)

**S1 Table. Questionnaire contents of the SQFFQ and the lifestyle questionnaire.**
(PDF)

**S2 Table. Instruments and analytical conditions.**
(PDF)

**S3 Table. List of the metabolites.**
(PDF)

**S4 Table. Summary of cross-validation analyses.**
(PDF)

**S5 Table. Data analysis results of PLS-R.**
(XLSX)

**S6 Table. Statistic summary of estimated food intakes in the original data.**
(XLSX)

**S7 Table. Statistic summary of metabolite concentrations in the original data.**
(XLSX)

## Acknowledgments

We thank the residents of Tsuruoka City for their interest in our study and the members of the Tsuruoka Metabolic Cohort Study team for their commitment to the project. Our special thanks go to Professor Yasunori Sato for his valuable suggestions.

## Author Contributions

**Conceptualization:** Eriko Shibutami, Toru Takebayashi.

**Data curation:** Sei Harada, Akiyoshi Hirayama, Masahiro Sugimoto, Toru Takebayashi.

**Formal analysis:** Eriko Shibutami.

**Funding acquisition:** Sei Harada, Tomoyoshi Soga, Masaru Tomita, Toru Takebayashi.

**Investigation:** Sei Harada, Ayako Kurihara, Kazuyo Kuwabara, Suzuka Kato, Miho Iida, Miki Akiyama, Daisuke Sugiyama, Masahiro Sugimoto, Toru Takebayashi.

**Methodology:** Eriko Shibutami, Toru Takebayashi.

**Project administration:** Sei Harada, Toru Takebayashi.

**Resources:** Akiyoshi Hirayama, Asako Sato, Kaori Amano, Tomoyoshi Soga, Masaru Tomita.

**Software:** Eriko Shibutami.

**Supervision:** Tomoyoshi Soga, Masaru Tomita, Toru Takebayashi.

**Validation:** Eriko Shibutami, Ryota Ishii.

**Visualization:** Eriko Shibutami.

**Writing – original draft:** Eriko Shibutami, Toru Takebayashi.

**Writing – review & editing:** Eriko Shibutami, Masahiro Sugimoto, Toru Takebayashi.

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
