## [Decision Letter · Decision Letter 0]

25 Nov 2020

PONE-D-20-30253

Charged metabolite biomarkers of food intake assessed via plasma metabolomics in a population-based observational study in Japan

PLOS ONE

Dear Dr. Takebayashi,

Thank you for submitting your manuscript to PLOS ONE. After careful consideration, we feel that it has merit but does not fully meet PLOS ONE’s publication criteria as it currently stands. Therefore, we invite you to submit a revised version of the manuscript that addresses the points raised during the review process.

We look forward to receiving your revised manuscript.

Kind regards,

Nurshad Ali

Academic Editor

PLOS ONE

Journal Requirements:

2. In your Methods section, please state the volume of the blood samples collected for use in your study.

Reviewers' comments:

Reviewer's Responses to Questions

**Comments to the Author**

1. Is the manuscript technically sound, and do the data support the conclusions?

Reviewer #1: Yes

Reviewer #2: Yes

2. Has the statistical analysis been performed appropriately and rigorously? 

Reviewer #1: Yes

Reviewer #2: Yes

3. Have the authors made all data underlying the findings in their manuscript fully available?

Reviewer #1: Yes

Reviewer #2: No

4. Is the manuscript presented in an intelligible fashion and written in standard English?

Reviewer #1: Yes

Reviewer #2: No

5. Review Comments to the Author

Reviewer #1: This study provides comprehensive analyses of the effects of dietary intake on plasma metabolite concentrations in a free-living Japanese population and potential candidate biomarkers of food consumption. The study was well-planned and carefully designed for both epidemiological and metabolomics analyses. The findings might help to frame references for considering the effects of dietary factors in further studies on biomarkers and nutrition profiling. Given that only a few large-scale epidemiological studies were conducted in Asian populations, this study provides valuable insight into Japanese regional habitual dietary characteristics and geographically-suitable candidate food biomarkers.

The title accurately reflects the content and, in general, the abstract presents an adequate synopsis of the paper.

The introduction provides a good, generalized background of the topic with logically organized, clear and well-argued narrative. The objective is clearly defined.

Key findings are well-summarized and arranged in a logical sequence that generally follows the methodology section. Non-textual elements, such as, figures and tables are self-explanatory and further illustrate the findings in an understandable manner. Please, correct the spelling for “cholesterol” in Table 1.

Strengths and limitations of the present research are well-articulated. Conclusions highlight main outcomes and future perspectives.

Reviewer #2: General comments

This is an interesting manuscript that describes a well conducted analysis involving a fairly large sample of participants within the Tsuruoka Metabolome Cohort Study relating a set of targeted metabolites to dietary variables expressing food intake with the objective of identifying metabolic signatures. The study is overall well conducted. There are three major largely improvable elements in the current version of the manuscript: first, the way several details are described in the text should enhance, please refer to the detailed comments; second, it is arguable whether the metabolic signatures that emerged during statistical analysis could claim to be defined as biomarker of intake. The discussion would benefit if a critical evaluation on whether these quantities could be used as objective measurements of dietary exposure(s) in etiological studies with respect to the risk of developing chronic conditions. Third, as opposed to biomarkers for specific food items, can we identify biomarkers of food groups? The question is legitimate, but deserves some discussion of important elements

Detailed comments

Abstract – line 27: the expression “Food intake biomarkers can be critical tools” is very vague. Food intake biomarker are objective measurements of intake. Also, why future nutrition studies? There is room to improve future and ongoing nutritional studies. Please consider revision.

Line 29: replace problems with conditions.

Line 30: replace surveys with assessments.

Line 35: replace statuses with status.

Line 35: replace diet with dietary.

Line 45-46: replace this result with these results. I suggest to emphasize the importance of generating objectives quantities to be used for nutritional profiling. Please consider revision.

Introduction, line 50: please describe what nutrition is rather than what it is not.

Line 51: replace problems with conditions.

Line 55: replace ‘random and systematic errors cannot be omitted as long as depend on self-reported surveys’ with ‘random and systematic affect self-reported …’

Line 56: replace surveys with assessments.

Line 58: the sentence ‘Metabolomics is a data-driven bio-scientific study’ expresses an unclear concept. Please revise it.

Line 62: the expression ‘profiling of epidemiological research findings’ is very unclear. Please avoid vague statements like this and consider revising it.

Line 62: It is arguable that metabolomics provides a tool to predict effects on disease biomarker. Why disease biomarkers?

Line 68: is there evidence that metabolite markers are reproducible? Please avoid use of vague sentences like this.

Line 70: replace targeting with targeted.

Line 71: replace targeting with targeted.

Lines 73-79. This sentence is probably more suitable for the discussion.

Line 74: population profiling of what?

Line 80-86: these are very general considerations. Move it to the discussion as well?

Line 111: replace Fig with Figure.

Line 118: is the second part of the sentence with what reported in the first part?

Line 122: “We subjected fasting plasma” – the meaning is unclear, please consider revision.

Line 130: “in a series of studies” – which studies, for which purposes?

Line 131: please provide a reference to the software.

Line 138-139: “The SFFQ was validated …. with correlation coefficients”. This information is insufficient. Correlation between what and what?

Line 142: This scale is only partially informative. Assessments of dietary items seemed very oriented to frequencies of intake per day. Could the Authors clarify how were food consumed once or twice per week?

Line 144-145: eventually it is not clear whether alcohol was assessed within the dietary or the lifestyle questionnaire. Please clarify.

Line 165: the percentage threshold used to exclude metabolites with values below LOD (90%) seems high indeed. May metabolites were not informative in PLSR analysis. How many metabolites would have been excluded with threshold of 50% or 70%?

Line 165: insert study between entire and population.

Line 169: “certain substances”. Which substances?

Line 169. Please spell out PLS-R.

Line 170-171: move sentence before the one starting in line 168.

Line 176: what about zero values?

Line 179-180: is the sentence “that involves a statistical model comparison for improving the goodness-of-fit” necessary or informative? Please consider revision.

Line 183: what is the difference between the explained variation and the predicted variation in Y?

Line 184: “Potential food intake marker”. Rather, the contribution of individual metabolites in the metabolic signature for each food group were evaluated.

Line 187: Please consider this description: with partial rank-order Spearman correlation coefficients, controlling for.

Line 190: except for outlier detection, what is JMP? And it seems still a SAS product, according to the reference.

All tables lack important details to appreciate what quantities are reported. For example, what are numbers in parenthesis in Table 2? If SD, please replace SD values with 10th – 90th ranges. In Table 3, as these are key quantities, report information in the footnote to make readers understand what Q^2_cum_c values are.

6. PLOS authors have the option to publish the peer review history of their article (what does this mean?). If published, this will include your full peer review and any attached files.

Reviewer #1: No

Reviewer #2: **Yes: **Pietro Ferrari

---

## [Author Response · Author response to Decision Letter 0]

3 Jan 2021

Response to Reviewers on "PLOS ONE Decision: Revision required [PONE-D-20-30253] - [EMID:06863db530440990]"

We greatly appreciate your efforts and your helpful comments in reviewing our article. We have incorporated all of your feedback in the revised manuscript. We have responded below in blue to your comments item-by-item. Changes are shown in the Revised Manuscript with Tracked Changes (the page and line numbers below are indicated the numbers when tracked changes are shown in the file). 

Journal Requirements:

Thank you for reminding us about the manuscript style. We have ensured that the manuscript meets PLOS ONE's style requirements.

2. In your Methods section, please state the volume of the blood samples collected for use in your study.

Thank you for pointing out the insufficient information. A 16 ml blood sample was collected from each participant and divided into 0.5–1 ml portions. The extracted metabolites were stored frozen until used for analysis. To clarify the sample collection method, we added these information to the Materials and methods. (Page 6, Lines 136-140)

Thank you for the comment. Because the FFQ is copyrighted by Nagoya City University, who is the developer of the questionnaire, we are unable to attach the original version here. Instead, we provided the summary of the questionnaire contents as Supporting Information (S1 Table), referring to their paper (Tokudome 2004, Reference #27), in which the detailed content of the questionnaire is shown, as well as providing more details of the assessment in the main text with references (Pages 6-7, Lines 151-162, References #26-#29). Please also see in the response to the Detail comments for Line 142 and Line 176 later.

4. We note that you have indicated that data from this study are available upon request. PLOS only allows data to be available upon request if there are legal or ethical restrictions on sharing data publicly. For information on unacceptable data access restrictions, please see 

http://journals.plos.org/plosone/s/data-availability#loc-unacceptable-data-access-restrictions.

Because there are ethical restrictions on sharing data publicly, the study prohibits any public data sharing, and original data from this study are only available upon request as follows:

Most relevant data are within the paper and its Supporting Information files. Raw data cannot be made publicly available, as study participants did not consent to have their information freely accessible. Based on this lack of consent, the Ethics Committee for Tsuruoka Metabolomics Cohort Study (which includes representatives of Tsuruoka citizens, administration of Tsuruoka City, a lawyer, and expert advisers) strictly prohibits any public data sharing because data contain potentially identifying or sensitive disease information. Data accession requests may be sent to the administration of the Ethics Committee for the Tsuruoka Metabolomics Cohort Study. The data will be shared after a review of the purpose and permission by the ethics committee. Contact information for the Ethics Committee for Tsuruoka Metabolomics Cohort Study is the administrator of the committee, Yutaka Sato, who may be contacted at the following email address: ytk.s@city.tsuruoka.yamagata.jp. Address: 9-25 Babacho, Tsuruoka City, 997-8601, Japan

We also refer to this information for data disclosure in the revised cover letter.

Reviewers' comments:

Reviewer #1: This study provides comprehensive analyses of the effects of dietary intake on plasma metabolite concentrations in a free-living Japanese population and potential candidate biomarkers of food consumption. The study was well-planned and carefully designed for both epidemiological and metabolomics analyses. The findings might help to frame references for considering the effects of dietary factors in further studies on biomarkers and nutrition profiling. Given that only a few large-scale epidemiological studies were conducted in Asian populations, this study provides valuable insight into Japanese regional habitual dietary characteristics and geographically-suitable candidate food biomarkers.

The title accurately reflects the content and, in general, the abstract presents an adequate synopsis of the paper. 

The introduction provides a good, generalized background of the topic with logically organized, clear and well-argued narrative. The objective is clearly defined.

Key findings are well-summarized and arranged in a logical sequence that generally follows the methodology section. Non-textual elements, such as, figures and tables are self-explanatory and further illustrate the findings in an understandable manner. Please, correct the spelling for “cholesterol” in Table 1. 

Strengths and limitations of the present research are well-articulated. Conclusions highlight main outcomes and future perspectives.

We are thankful for the time and energy you expended as well as for providing thoughtful comments. We are hopeful that our findings will create more opportunities for metabolic profiling in nutritional studies through the new dietary assessment tools, while simultaneously bringing consideration to the influence of dietary factors in exploring biomarkers in various fields. Finally, thank you for pointing out the typo. We corrected "choresterol" to "cholesterol" (Table 1: Page 10, 241).

Reviewer #2: General comments

This is an interesting manuscript that describes a well conducted analysis involving a fairly large sample of participants within the Tsuruoka Metabolome Cohort Study relating a set of targeted metabolites to dietary variables expressing food intake with the objective of identifying metabolic signatures. The study is overall well conducted. There are three major largely improvable elements in the current version of the manuscript: first, the way several details are described in the text should enhance, please refer to the detailed comments; second, it is arguable whether the metabolic signatures that emerged during statistical analysis could claim to be defined as biomarker of intake. The discussion would benefit if a critical evaluation on whether these quantities could be used as objective measurements of dietary exposure(s) in etiological studies with respect to the risk of developing chronic conditions. Third, as opposed to biomarkers for specific food items, can we identify biomarkers of food groups? The question is legitimate, but deserves some discussion of important elements.

We appreciate the time and effort you have given and for providing insightful feedback on ways to strengthen our paper. Firstly, we apologize for our insufficient description of the original manuscript as you pointed out. We carefully examined and revised the manuscript according to your suggestions. Please see the responses in the Detailed comments below. 

Secondly, thank you for providing thoughtful insights related to the effectiveness of the metabolic signatures. We agree that among various types of dietary biomarkers defined previously (Scalbert 2014, Reference #8; Jenab 2009, #9), it is arguable whether the metabolite signatures we observed could be defined as biomarkers of intake (i.e., the concentration of replacement dietary biomarker). Follow-up surveys are ongoing now to estimate diet-disease risk association as well as to examine the function, mechanism of action, and the validity and reproducibility of the objective quantities. Effects of endogenous and extrinsic factors on metabolite concentrations are also to be evaluated. So, we added these arguments to the Strengths and limitations (Page 23, Lines 511-517). 

Finally, you have asked an reasonable question. An assessment categorized by food groups is likely to make it more challenging to distinguish the effects of specific food items that may differentially associate with metabolites, but it may also improve the interpretability of dietary status based on composite intake of various food items. Thus, we can say it is more suitable for practical applications to assess long-term dietary habits. Indeed, the approach by food groups has been adopted to identify habitual food intake biomarkers in typical epidemiological studies (e.g., Playdon 2017, Reference #14; Gorska-Warsewicz 2017, #64; Zheng 2014, #71). So we added these aspects to the Strengths and limitations as well (Pages 22-23, Lines 499-504).

Detailed comments

Abstract – line 27: the expression “Food intake biomarkers can be critical tools” is very vague. Food intake biomarker are objective measurements of intake. Also, why future nutrition studies? There is room to improve future and ongoing nutritional studies. Please consider revision.

Thank you for pointing out the vague description. We tried to mention the future possibility of replacing the current self-reported assessment tools which are widely conducted, such as Dietary-Records, 24-Recall, and FFQs; however, this statement was not accurate. Therefore, we revised the description as follows:

"Food intake biomarkers can be critical tools for future nutrition studies to deliver objective ways to assess dietary exposure." → "Food intake biomarkers are critical tools that can be used to objectively assess dietary exposure for both epidemiological and clinical nutrition studies." (Page 2, Lines 27-29)

Line 29: replace problems with conditions.

Thank you for your suggestion. As per your feedback, we revised the wording as follows:

"problems" → "conditions" (Page 2, Line 30) 

Line 30: replace surveys with assessments.

As per your feedback, we revised the wording as follows:

"surveys" → "assessments" (Page 2, Line 32) 

Line 35: replace statuses with status.

Line 35: replace diet with dietary.

As per your feedback, we revised the sentence as follows:

"their health statuses and diet intake" → "patients’ health status and dietary intake" (Page 2, Lines 36-37)　　 

Line 45-46: replace this result with these results. I suggest to emphasize the importance of generating objectives quantities to be used for nutritional profiling. Please consider revision.

Thank you for your feedback. We agree to emphasize the importance of generating objectives quantities, so we revised descriptions as follows:

[Abstract]

"This result will open the way for the application of simple and objective new dietary assessment tools for profiling in nutritional studies." → "These results will open the way for the application of new reliable dietary assessment tools not by self-reported measurements but through objective quantification of biofluids." (Page 2, Lines 47-50)

[Introduction]

"evaluating food intake" → "objective food intake evaluations" (Page 3, Line 69)

"for new reliable dietary assessment tools by objective quantification of biofluids." (Added to Page 4, Line 99)

[Conclusions]

"This result will open the way for the application of simple and objective new dietary assessment tools for profiling in nutritional studies." → "These results will open the way for the application of new reliable dietary assessment tools by objective quantification of biofluids." (Page 24, Lines 529-531)

Introduction, line 50: please describe what nutrition is rather than what it is not.

Line 51: replace problems with conditions.

Thank you for your helpful consideration. We revised the description as follows:

"The role of nutrition studies is not to simply clarify individual or group food intake, but to unravelassociations between dietary exposure and specific health problems." → "Nutrition studies aim to reveal associations between dietary exposure and specific health conditions by clarifying individual or group food intake." (Page 3, Lines 54-56)

As per your feedback, we revised the wording in the above sentence as follows:

"problems with" → "conditions by"

Line 55: replace ‘random and systematic errors cannot be omitted as long as depend on self-reported surveys’ with ‘random and systematic affect self-reported …’

Line 56: replace surveys with assessments.

Thank you for your helpful suggestion. We revised the sentence as follows:

"random and systematic errors cannot be omitted as long as depend on self-reported surveys" → "random and systematic errors affect self-reported assessments" (Page 2, Lines 31-32; Page 3, Lines 60-61)

As per your feedback, we revised the wording in the above sentence as follows:

"surveys" → "assessments" 

Line 58: the sentence ‘Metabolomics is a data-driven bio-scientific study’ expresses an unclear concept. Please revise it.

Thank you for pointing out the vague sentence. We revised the description as follows:

"Metabolomics is a data-driven bio-scientific study, wherein a large amount of data is collected from biochemical samples " → "Metabolomics is one of the core subject fields of systems biology, wherein comprehensive data of all measurable metabolite concentrations are collected from biochemical samples " (Page 3, Lines 64-66)

Line 62: the expression ‘profiling of epidemiological research findings’ is very unclear. Please avoid vague statements like this and consider revising it.

Thank you for pointing out the unclear description. We revised the description as follows:

"profiling of epidemiological research findings" → "response to nutritional modulations in observational and interventional studies" (Page 3, Lines 69-71)

Line 62: It is arguable that metabolomics provides a tool to predict effects on disease biomarker. Why disease biomarkers?

We agree with you that the description was vague, so we revised the description as follows:

"predicting effects on disease biomarkers" → "metabolic profiles as biological consequences of dietary intake" (Page 3, Lines 71-72)

Line 68: is there evidence that metabolite markers are reproducible? Please avoid use of vague sentences like this.

Thank you again for pointing out the unclear description. We revised it as follows:

"Thus, it is feasible to identify such food-specific and reproducible metabolite markers" → "Thus, we can expect to identify such food-specific metabolite markers" (Page 4, Lines 77-78)

Line 70: replace targeting with targeted.

Line 71: replace targeting with targeted.

As per your feedback, we revised the wording as follows:

"targeting" → "targeted" (Page 4, Line 80) 

"non-targeting" → "non-targeted" (Page 4, Lines 80-81) 

Line 73-79. This sentence is probably more suitable for the discussion.

Line 74: population profiling of what?

Line 80-86: these are very general considerations. Move it to the discussion as well?

We agree with you that these descriptions should be appropriate for the Discussion. Therefore, we moved these perspectives to the Discussion whereas a brief description was left in the Introduction as follows:

[Remains in the Introduction]

"Although a considerable number of attempts …… characteristics of free-living individuals is expected." (Page 4, Lines 79-96)

[Moved to the Discussion]

"A lot of pioneering efforts of dietary biomarkers …… or were mostly implemented among Western populations." (Page 16, Lines 329-343)

Thank you for pointing out the vague description. We revised it in the above paragraph as follows:

"population profiling" → "dietary profiling in epidemiological studies" (Page 16, Lines 331-332)

Line 111: replace Fig with Figure.

As per your feedback, we revised it as follows: 

"Fig → "Figure" (Page 5, Line 122)　

Line 118: is the second part of the sentence with what reported in the first part?

Thank you for pointing out the inconsistent description. The second part was "the quantity of alcohol consumed per occasion." However, the sentence was unnecessary here, so we omitted it (Page 6, Lines 128-129). Please also see the response to the comment for Line 144-145 later. 

Line 122: “We subjected fasting plasma” – the meaning is unclear, please consider revision.

Thank you for pointing out the vague use of words. We revised it as follows: 

"We subjected fasting plasma samples to a non-targeted metabolome analysis" → "The fasting plasma samples were analyzed to obtain non-targeted metabolomics data" (Page 6, Lines 133-134)

In addition, to clarify the sampling method, we additionally replaced words as follows:

"plasma samples" → "blood samples" (Page 5, Line 123)

Line 130: “in a series of studies” – which studies, for which purposes?

Thank you for pointing out the vague description. We described "in a series of studies" for the entity studies of the Tsuruoka Metabolome Study. However, the description was unnecessary, so we omitted the phrase. (Page 6, Line 143)

Line 131: please provide a reference to the software.

Thank you for your feedback. We provided a reference to the software as follows: 

"our proprietary software" → "our proprietary software, MasterHands [Sugimono 2010, Reference #25]" (Page 6, Line 144) 

Line 138-139: “The SFFQ was validated …. with correlation coefficients”. This information is insufficient. Correlation between what and what?

We apologize for providing insufficient information. Actually, the validity and reproducibility of the SQFFQ had been assessed with correlation coefficients between the SQFFQ and 28-day weighted diet records as well as between two SQFFQs administered at a one-year interval. So, we revised the information to be simpler with references as follows:

"The SQFFQ was validated for assessing energy, ……, and food consumption with correlation coefficients" → "The validity and reproducibility of the SQFFQ had been assessed for energy, ……, and food consumption [Reference # 28, #29]." (Pages 6-7, Lines 151-153) 

In addition, we revised the following abbreviation as per the original articles of the FFQ.

"SFFQ" → "SQFFQ" (All such abbreviations)

Line 142: This scale is only partially informative. Assessments of dietary items seemed very oriented to frequencies of intake per day. Could the Authors clarify how were food consumed once or twice per week?

We apologize for providing inadequate information. We had asked about the frequency of intake over a wider time range than per day, so we provided detailed questionnaire information as follows:

"Responses to the questions on food intake were categorized at eight levels: from "never or rarely" to "3+ per day"." → "Responses to the questions on food intake were categorized at eight levels (never or seldom, 1 to 3 times per month, 1 to 2 times per week, 3 to 4 times per week, 5 to 6 times per week, once per day, twice per day, and three times or more per day) [Reference #27]." (Page 7, Lines 155-157)

"cups → "portions (cups/pieces) " (Page 7, Line 159)

Line 144-145: eventually it is not clear whether alcohol was assessed within the dietary or the lifestyle questionnaire. Please clarify.

Thank you for pointing out the unclear description. To clarify the information that alcohol consumption was assessed within the lifestyle questionnaire, separate from the FFQ, we revised it as follows: 

"Alcohol intake was calculated based on the reported number of drinks per week, as well as the frequency of alcohol consumption." (Deleted, Page 6, Lines 128-129)

"the frequency of intake of 47 food items and different kinds of alcohol" → "the frequency of intake of 47 food items" (Page 7, Line 154)

"the number of drinks and the frequency of drinking in a questionnaire on lifestyle administered simultaneously." → "different kinds of alcohol, the number of drinking days per month/week, and the number of drinks per occasion in a questionnaire on lifestyle (see more details of questionnaires in the S1 Table)." (Page 7, Lines 160-162)

Line 165: the percentage threshold used to exclude metabolites with values below LOD (90%) seems high indeed. May metabolites were not informative in PLSR analysis. How many metabolites would have been excluded with threshold of 50% or 70%?

Thank you for confirming the threshold of the LOD. We had initially set a threshold of 90% (Fukai2016, Iida 2016). However, after checking the final values as per your feedback, the lowest detection rate in the final 94 metabolites included in the analysis with the threshold was eventually 41% (trigonelline). In other words, metabolites below the LOD in more than 60% (that is, less than 40% of the detection rate) had been excluded indeed. Incidentally, if the threshold were 50%, 91 metabolites would be included. Therefore, we think the PLS analysis of the 94 metabolites was appropriate and revised the information as follows: 

"in more than 90% of" → "in more than 60% of" (Page 8, Line 186)

Line 165: insert study between entire and population.

As per your feedback, we revised it as follows:

"entire population" → "entire study population" (Page 8, Line 186)

Line 169: “certain substances”. Which substances?

Thank you for pointing out the vague use of words. We included the description as follows: 

"certain substances" → "substances which might change together due to biochemical interactions in vivo" (Page 8, Lines 192-193)

Line 169. Please spell out PLS-R.

Thank you for confirming the spelling of the word. The PLS-R had been spelled out in the first appearance in the Introduction (Page 5, Line 101) according to the manuscript guidelines, so we have left it as was (Page 8, Line 196).

Line 170-171: move sentence before the one starting in line 168.

We agree with you that the sentence should be moved to the beginning of the paragraph, so we moved the sentence there. (Page 8, Lines 190-191)

Line 176: what about zero values?

Thank you for confirming the missing values. If the "zero values" you are referring to is where the frequency level is 0 (never or seldom), then they were not included in missing values as we provided minimum intake values by a conversion weight assigned of 0.05 for the food intake frequency per day for them. To clarify the estimate calculation, we added the information in the Dietary assessment as follows: 

"If the intake frequency was less than once per day, a conversion weight was assigned (never or seldom: 0.05, 1 to 3 times per month: 0.1, 1 to 2 times per week: 0.2, 3 to 4 times per week: 0.5, and 5 to 6 times per week: 0.8). Alcohol intake was calculated based on the reported frequency and quantity consumed per occasion." (Added to Page 7, Lines 170-173)

Line 179-180: is the sentence “that involves a statistical model comparison for improving the goodness-of-fit” necessary or informative? Please consider revision.

We agree with you that the sentence was unnecessary so we omitted the phrase. (Page 9, Lines 203-204)

Line 183: what is the difference between the explained variation and the predicted variation in Y?

Thank you for your question. To clarify the statistical terms, we added a brief description in the Discussion as follows:

"and the predicted variation in the Y matrix (Q2) " → "the predicted variation in the Y matrix (Q2), and their cumulative values" (Page 9, Lines 208)

"In a PLS-R model, R2Y is the proportion of variance in the dependent factors that is predictable from the independent factors, while Q2 is the R2 when the model built on the training set is applied to the test set. Adding a factor always raises R2Y, whereas Q2 does not raise in case of over-fitting. Therefore, the closer the cumulative Q2 is to 1, the better the predictive performance of the model." (Page 9, Lines 208-212)

Line 184: “Potential food intake marker”. Rather, the contribution of individual metabolites in the metabolic signature for each food group were evaluated.

Thank you for the helpful suggestion. We revised it as follows: 

"Potential food intake marker" → "The contribution of individual metabolites in the metabolic signature for each food group" (Page 9, Lines 212-214)

Line 187: Please consider this description: with partial rank-order Spearman correlation coefficients, controlling for.

We agree with your suggestion and revised it as follows: 

"via Spearman’s rank-order correlation analysis with partial correlation coefficients, excluding the effects of " → "with partial rank-order Spearman correlation coefficients, controlling for" (Page 9, Lines 216-218)

[Fig 2 legend] (page 15, Lines 323-324)

"Partial correlation coefficients were obtained by Spearman’s rank-order method, excluding the effect of potential confounding for" → "This heatmap was generated by partial rank-order Spearman correlation coefficients, controlling for" (Page 16, Lines 323-325) 

Line 190: except for outlier detection, what is JMP? And it seems still a SAS product, according to the reference.

Thank you for asking for the applications. Although both software products are provided by SAS Institute, JMP is a visual analysis and statistics software with a wizard-based user interface through its proprietary scripting language, JSL, while SAS is a programming software through the SAS language. As JMP has various visualization functions, we used the JMP for the outlier detection to visualize the results effectively. 

As related to the software, we added the information in the text as follows:

"All statistical analyses were performed using SAS version 9.4 (SAS Institute Inc., Cary, NC, USA), except for outlier detection, which was performed using JMP version 15 (SAS Institute Inc., Cary, NC, USA). " → "Statistical analyses were performed using SAS version 9.4 (SAS Institute Inc., Cary, NC, USA), and JMP version 15 (SAS Institute Inc., Cary, NC, USA) was used for outlier detection to visualize the results." (Page 9, Lines 218-221)

All tables lack important details to appreciate what quantities are reported. For example, what are numbers in parenthesis in Table 2? If SD, please replace SD values with 10th – 90th ranges. In Table 3, as these are key quantities, report information in the footnote to make readers understand what Q^2_cum_c values are.

We apologize for providing insufficient information in the Tables, so we revised important information on the footnotes as follows:

[Table 1] (Pages 10-11, Lines 243-247)

BMI, body mass index. 

a Mean, standard deviation in parentheses (all such values).

b Median, 25th-75th percentiles in parentheses (all such values).

c Percentage for categorical variables (all such values).

d Values are shown as Ethanol equivalent.

[Table 2] (Page 11, Lines 251-254)

FFQ, food frequency questionnaire.

a Values are presented as mean and 10th-90th percentiles in parentheses.

b Values are calculated according to the percentage of ethanol and shown in comparison to sake. 

(In the main text)

" SDs" → "10th - 90th percentile ranges" (Page 8, Line 184)

[Table 3] (Pages 13, Lines 302-311)

a ……

b ……

c Cumulative predicted variation in the Y matrix for optimal factor numbers, calculated as 1 - (the cumulative predicted residual sum of squares / the cumulative sum of squares). The value indicates the predictive performance of the model. For cases with an optimal factor number of less than two, the factor number was set to two and the result was shown in parentheses.

d Partial rank-order Spearman's correlation coefficients between food consumption and metabolite concentration, controlling for sex, smoking, and physical activity levels. 

e ……

[S3 Table] 

PRESS, predicted residual sum of squares.

a Smallest number of factor numbers provided by Van der Voet test with a T2 critical value of p > 0.10. For cases with an optimal factor number of less than two, the factor number was set to two and the result was shown in parentheses.

b Cumulative explained variation in the X matrix. 

c Cumulative explained variation in the Y matrix.

d Cumulative predicted variation in the Y matrix.

e Data of male drinkers were used in the analysis.

References (Pages 24-, Lines 540-)

(Added)

(#9) Jenab M, et al. Hum Genet. 2009

(#25) Sugimoto M, et al. Metabolomics. 2010

(#27) Tokudome S, et al. APJCP. 2004

(#38) Abid Z, The Am J Clin Nutr. 2014

(#39) Wolk A. J Intern Med. 2017

List of the Supporting Information (at the end of the manuscript file)

"S1 Table. Questionnaire contents of the SQFFQ and the Lifestyle Questionnaire." (Added as per the Journal Requirements #3)

"S1 Table. Instruments …" → "S2 Table. Instruments …"

"S2 Table. List …" → "S3 Table. List …"

"S3 Table. Summary …" → "S4 Table. Summary …"

"S4 Table. Data analysis results." → "S5 Table. Data analysis results of PLS-R."

"S5 Table. Statistic …" → "S6 Table. Statistic …"

"S6 Table. Statistic …" → "S7 Table. Statistic …"

The table numbers in the main text and the files have been revised to match the above.

Other revisions

Finally, to provide accurate information, we revised the descriptions in the main text and tables as follows: 

[In the main text]

"three fit levels" → "three predictive performance levels" (Page 12, Line 261)

"(see more details of the results in the S5 Table)." (Added to Page 13, Lines 292-293)

"Meat intake biomarkers are important because meat consumption is likely to be associated with the risk of various cancers and chronic diseases [Abid 2014, Reference #38; Wolk 2017, #39]." (Added to Page 17, Lines 357-358)

"mandarin was" → "citrus fruits including mandarin were" (Page 20, Lines 446-447)

"metabolites" → "metabolites with high VIP scores" (Page 21, Line 470)

[Table 2] (Page 11, Line 249) / [S6 Table]

"Mandarin" → "Mandarin/orange/grapefruit"

"other fruit" → "other fruits"

[S2 Table] 

"Ref.) Hirayama A, Sugimoto M, Suzuki A, Hatakeyama Y, Enomoto A, Harada S, et al. Effects of processing and storage conditions on charged metabolomic profiles in blood. Electrophoresis. 2015. " (Added to the footnote) 

[S3 Table] / [S5 Table] / [S7 Table]

"aspartic acid" → "aspartate"

[S5 Table]

"SDMA" → "Symmetric Dimethylarginine"

"ADMA" → "Asymmetric Dimethylarginine"

"CSSG" → "Cysteineglutathione disulfide"

---

## [Decision Letter · Decision Letter 1]

20 Jan 2021

Charged metabolite biomarkers of food intake assessed via plasma metabolomics in a population-based observational study in Japan

PONE-D-20-30253R1

Dear Dr. Takebayashi,

We’re pleased to inform you that your manuscript has been judged scientifically suitable for publication and will be formally accepted for publication once it meets all outstanding technical requirements.

Kind regards,

Nurshad Ali

Academic Editor

PLOS ONE

Reviewers' comments:

Reviewer's Responses to Questions

**Comments to the Author**

1. If the authors have adequately addressed your comments raised in a previous round of review and you feel that this manuscript is now acceptable for publication, you may indicate that here to bypass the “Comments to the Author” section, enter your conflict of interest statement in the “Confidential to Editor” section, and submit your "Accept" recommendation.

Reviewer #1: All comments have been addressed

2. Is the manuscript technically sound, and do the data support the conclusions?

Reviewer #1: Yes

3. Has the statistical analysis been performed appropriately and rigorously? 

Reviewer #1: N/A

4. Have the authors made all data underlying the findings in their manuscript fully available?

Reviewer #1: Yes

5. Is the manuscript presented in an intelligible fashion and written in standard English?

Reviewer #1: Yes

6. Review Comments to the Author

Reviewer #1: I thank the authors for their very detailed replies to comments on the submitted version. Congratulations for well-designed study and valuable insight into Japanese regional habitual dietary characteristics and geographically-suitable candidate food biomarkers.

7. PLOS authors have the option to publish the peer review history of their article (what does this mean?). If published, this will include your full peer review and any attached files.

Reviewer #1: No

---

## [Editor Report · Acceptance letter]

25 Jan 2021

PONE-D-20-30253R1 

Charged metabolite biomarkers of food intake assessed via plasma metabolomics in a population-based observational study in Japan 

Dear Dr. Takebayashi:

I'm pleased to inform you that your manuscript has been deemed suitable for publication in PLOS ONE. Congratulations! Your manuscript is now with our production department. 

Kind regards, 

on behalf of

Dr. Nurshad Ali 

Academic Editor

PLOS ONE